# Copper Filled Poly(Acrylonitrile-co-Butadiene-co-Styrene) Composites for Laser-Assisted Selective Metallization

**DOI:** 10.3390/ma13102224

**Published:** 2020-05-12

**Authors:** Piotr Rytlewski, Bartłomiej Jagodziński, Tomasz Karasiewicz, Piotr Augustyn, Daniel Kaczor, Rafał Malinowski, Krzysztof Szabliński, Marcin Mazurkiewicz, Krzysztof Moraczewski

**Affiliations:** 1Department of Materials Engineering, Kazimierz Wielki University, 85-064 Bydgoszcz, Poland; bar.jag@ukw.edu.pl (B.J.); tomakara@ukw.edu.pl (T.K.); augustyn@ukw.edu.pl (P.A.); krzysztofszablinski1@gmail.com (K.S.); mazurkiewiczmarcin95@gmail.com (M.M.); kmm@ukw.edu.pl (K.M.); 2Łukasiewicz Research Network - Institute for Engineering of Polymer Materials and Dyes, 87-100 Toruń, Poland; d.kaczor@impib.pl (D.K.); ipts013@impib.pl (R.M.)

**Keywords:** infrared lasers, polymer composites, copper microspheres, surface activation, electroless metallization

## Abstract

Selective metallization of polymeric materials using the technique known as laser direct structuring (LDS) is intensively developed. In this technique, metallized products can be manufactured by injection molding or by 3D printing process if rapid prototyping is need. Special additives present in the polymer matrix enable direct electroless metallization only on the surface which was laser activated. This paper presents the results of using copper microparticles introduced into the poly(acrylonitrile-butadiene-styrene) (ABS) matrix at various amounts (up to about 5 vol %). ABS was selected due to its good processing and mechanical properties and as one of the most common thermoplastics used in 3D printing. The influence of copper on structural, mechanical, and processing properties as well as on the effects of laser surface activation were determined. Two types of infrared lasers were tested for surface activation: Nd:YAG fiber laser (λ = 1064 nm) and CO_2_ laser (λ = 10.6 µm). Various irradiation parameters (power, scanning speed, and frequency) were applied to find suitable conditions for laser surface activation and electroless metallization. It was found that the composites tested can be effectively metallized using the Nd:YAG laser, but only in a narrow range of radiation parameters. Activation with CO_2_ laser failed, regardless of applied irradiation conditions. It resulted from the fact that ablation rate and thickness of modified surface layer for CO_2_ were lower than for Nd:YAG laser using the same irradiation parameters (power, speed, and frequency of laser beams), thus the laser wavelength was crucial for successful surface activation.

## 1. Introduction

Thermoplastic functional composites are of great interest due to the ease of forming 3D products with complex shapes using mass-scale injection molding process or 3D printing process if rapid prototyping is necessary. Various functional properties can be incorporated in the molded parts by introducing special additives into the polymer matrix. When using conductive filler magnetic, antistatic, or electro-shielding properties of molded parts can be obtained which become more and more important in highly integrated advanced products [1]. In automotive, aircraft, medical, and mechatronic devices there is increasing need to integrate mechanical with electric functions by incorporating conductive truck onto the surface of constructional parts. This is realized by molded interconnect devices (MID) technology [2,3]. These devices can be perceived as three-dimensional rigid printed circuit boards (PCBs), which allow the optimal use of the installation space due to their three-dimensionality. Compared to standard epoxy resin PCBs, MID are more recyclable and their manufacturing requires less processing steps [4,5].

There are various approaches to metallize selectively 3D thermoplastic parts. Generally, it is realized by electroless metallization, which, however, requires prior activation of the polymer surface [6,7]. Surface is activated when seeded with catalytic species like various kind of metals able to reduce metallic ions from metallization bath [8,9,10]. In electroless copper plating, palladium is the most common catalyzer seeded by various chemical treatments on the polymer surface [11]. Special masks have to be involved in chemical activation methods to get selective activation and metallization. Beside chemical treatments, lasers became frequently applied to activate the surface selectively thus simplifying selective metallization. Laser irradiation can be used to activate the surface of the polymer material immersed in special solutions containing compounds of seeded metal as a catalyzer or to preactivate the surface which, in the next step, can be activated selectively by means of chemical treatment [12,13,14]. However, the most commonly used manufacturing method for 3D-MIDs is the laser direct structuring (LDS) method, in which special metal–organic compounds are co-compounded with the polymer material intended to be injection molded [15,16]. As a result of laser irradiation, organic ligands of the metalorganic compounds along with polymer matrix are ablated while heavier metallic atoms left on the surface, thus constituting active sites for reduction of metal ions from metallization bath. The commercial metal–organic fillers are ‘know-how’ of the leading producers, thus are publically unpublished. On the other hand, some of the researchers reported successful applications of organic as well as inorganic compounds, such as copper or nickel acetates or acetylacetonates and copper L-tyrosine [17,18,19,20,21,22,23,24], copper chromium oxides, hydroxides, or hydroxide phosphate [25,26,27,28].

In this work, possible application of microscopic copper particles as a metallization precursor for poly(acrylonitrile-butadiene-styrene) (ABS) for LDS technique was evaluated. One can expect that using this type of filler laser irradiation will uncovered copper particles, and thus create activated surface for direct electroless metallization. It can be predicated that copper particles will be less densely located on the surface as compared with nanoscopic metalorganic additives which are densely dispersed in polymer matrix. Therefore, the possibility to obtain electrolessly deposited continuous copper layer was the main objective of this study. On the other hand, assuming that copper filler can affect various structural properties of ABS subjected to thermal processing, much attention was also paid to these effects. Due to the best authors’ knowledge and broad literature review this type of filler used as a metallization precursor for LDS technique was not the subject of previous publications. Additional novelty of this study is reflected by application of the two types of infrared lasers (1064 nm and 10.6 µm) for which activation effects were compared.

## 2. Materials and Methods

### 2.1. Materials

The following materials were applied to produce composite materials and for electroless metallization process:poly(acrylonitrile-co-butadiene-co-styrene) (ABS) Terluran GP-35 Natural with a density of 1.04 g/cm^3^ and a flow rate of 55 cm^3^/10 min (10 kg, 220 °C) (Styrolution GmbH, Ludwigshafen, Germany);Copper (Cu) in the form of spherical particles (powder) with a purity of 98%, grain size 10–25 μm, density 8,96 g/cm³, melting point 1083.4 °C (Sigma - Aldrich, Poznań, Poland);Six components commercial electroless copper bath type M-Copper 85 (MacDermid-Poland, Łysomice, Poland) with formaldehyde 36% (POCH, Gliwice, Poland) as a reducing agent.

### 2.2. Processing

The composites made of ABS and copper powder as a filler were prepared by double-screw extrusion along with granulation process. The content of copper powder associated with designation of composite samples are listed in Table 1. Before extrusion, ABS and copper powder were separately dried at 80 °C for 24 h. Extrusion was performed using co-rotating double-screw extruder type BTSK 20/40D (Bűhler, Braunschweig, Germany) with maintaining following temperatures of extruder zones: 210, 215, 220, 225, and 220 °C of the extruder head. A three-holes extruder head was used to form filaments which were cooled by airflow and then in-line granulated by rotating knives.

The obtained composite granules were next injection molded to form the samples (plates) for laser surface activation and electroless metallization. It was performed using injection-molding machine type Tederic TRX 80 ECO 60 (Tederic Machinery Manufacture, Hangzhou, China) with the temperatures of barrel heating zones set to 220, 220, 215, 225, and 60 °C of the mold. The dimensions of the injection molded plates were 60 × 60 × 1 mm.

The composite plates were laser irradiated in order to eject ABS matrix (laser ablation) from the composites surface layer and uncover copper particles, which can act as catalytic centers for the reduction of copper ions from metallization bath. Two types of lasers were tested: Nd:YAG fiber laser (λ = 1064 nm, P_max_ = 20 W) type TS-20W (Techsol, Bielsko-Biała, Poland) and CO_2_ laser (λ = 10.6 um, P_max_ = 60 W) type 900N (Techsol, Bielsko-Biała, Poland). The preliminary range of irradiation parameters was set by varying power, frequency, and speed of the laser beam, for which the metallization effects (deposited copper layer) was detected.

After laser irradiation samples were immersed in metallization bath for 60 min. The bath was prepared according to the manufacturer’s instructions. Its temperature was kept constant at 46 °C and was constantly aerated.

### 2.3. Measurements

Thermal stability of the samples were evaluated by means of thermogravimetric analysis (TGA), oxidation induction time (isothermal OIT) and oxidation induction temperature (dynamic OIT*). For TGA analysis samples of the mass 8.1 ± 0.3 mg was placed in platinum pan and heated from room temperature up to 600 °C at a constant heating rate of 10 °C/min under constant nitrogen flow. The OIT and OIT* measurements were performed using differential scanning calorimeter Q200 (TA Instruments, New Castle, DE, USA). The experimental procedure to determine OIT consisted of heating the samples at rate of 10 °C/min from 40 to 150 °C; isothermal storing for 3 min and then N_2_ to O_2_ switch (gas flow rate 50 cm^3^/min) while recording heat flow of the samples. Dynamic OIT* was determined as onset temperature for increase in heat flow measured for the samples heated from 50 to 280 °C in an oxygen atmosphere. Both types of OIT/OIT* values were determined according to ISO standard [29].

The composite samples were also tested by means of standard differential scanning calorimetry (DSC) using the same calorimeter as for OIT measurements. The samples of the mass 2.1 ± 0.1 mg was placed in aluminum pan, and were subjected to heating/cooling/heating cycles at constant rate of 20 °C/min under nitrogen flow with lower and upper temperature limits −70 °C and 250 °C, respectively.

The measurements of volume resistivity (R_v_) was performed with Model 8009 electrodes and model 6517A electrometer (Keithley Instruments Inc., Solon, OH, USA), according to the ASTM D256-07. The examinations were carried out using a constant voltage of 100 V and the measurement time before a change in the voltage polarization was 30 s (polarization period). The assumed current values for individual samples are arithmetic means of ten measurements acquired at the end of each polarization periods.

The tensile tests were performed with a tensile testing machine, type Instron 3367 (Instron, Norwood, MA, USA), for determination of tensile strength (σ_M_), tensile stress at break (σ_B_), tensile strain at tensile strength (ε_M_), tensile strain at break (ε_B_), Young’s modulus (E), and break energy (E_B_). The tests were performed in accordance with an appropriate standard [30,31]. The parameters E, σ_M_, σ_B_, ε_M_, ε_B_, and E_B_ were determined using 12 individual samples. The final values of these quantities were derived as arithmetic means of 10 results, two extreme ones being neglected.

The measurements of MFR were performed using an MP 600 plastometer (Tinius Olsen, Horsham, PA, USA) due to the procedure specified in an appropriate standard [32]. For all the samples, the measuring temperature was 200 °C and the piston load, 5 kg. The MFR was determined using 12 individual samples. The final value of MFR was derived as an arithmetic mean of 10 results, two extreme ones being neglected.

Effects of laser irradiation and electroless metallization were evaluated based on optical and scanning electron microscopies (SEM). The optical images were taken by microscope DMS300 (Leica, Wetzlar, Germany) whereas SEM measurement were performed using microscope SU 8010 (HITACHI, Tokyo, Japan). For SEM measurements samples were coated with thin evaporated gold layer to record high resolution surface topography.

Possible changes induced by laser irradiation in chemical structure of the composites were evaluated based on energy-dispersive X-ray (EDX). The elemental surface layer composition was performed using the SEM microscope additionally equipped with energy-dispersive X-ray (EDX).

## 3. Results and Discussion

### 3.1. Characterization of the Composites

The ABS copolymer has complex polymer chain structure consisting of a free styrene-acrylonitrile (SAN) copolymer with butadiene (B) grafted blocks. Macroscopically, SAN constitutes a continuous phase in which PB phase is dispersed. The PB phase contributes to the toughness of this copolymer even at low temperatures, while nitrile groups from neighboring chains, being polar, attract each other and bind the chains together, making ABS stronger than pure polystyrene. High impact strength and elastic modulus, ease of processing, and common applications in electroless metallization and 3D additive manufacturing motivated the use of ABS as the matrix of composites intended for laser-assisted selective metallization and 3D printing applications. On the other hand, high susceptibility to thermal, oxidative, and/or ultraviolet-induced degradations can be identified as disadvantageous. 

Copper applied in this work was in the form of powder with particles of irregular spherical shapes with diameters ranging from about 3 to 20 µm (Figure 1).

These particles size were generally in accordance with that declared by the supplier (about 20 µm). It is expected that copper can catalyze degradation processes in some thermoplastic polymers. Although many of the studies concerned degradation of neat ABS, the possible influence of copper filler on processing-induced degradation of ABS was not broadly discussed and therefore attention was also focused on that aspect in this work.

Initially, DSC tests were carried out to determine the possible effect of copper on the ABS structure. Due to the styrene (S), acrylonitrile (AN), and B blocks, three characteristic phase transitions can be expected. The glass transition of B phase (at about −53 °C [33] or even at about −80 °C [34]) is difficult to be detect by DSC, even using high heating/cooling rates. In this study heating/cooling rate was 20 °C/min but glass transition for B phase was not evidenced. The glass transition of S block at about 109 °C and melting of AN crystallites at about 140 °C were identified (Figure 2).

The curves shown in Figure 2 are obtained for the second heating cycle to compare the effect of copper on the samples structure formed under the same thermal conditions (thermal history). Based on these curves, the combined enthalpy (H) of the glass transition for S and melting of the crystalline AN phase was calculated in total for the temperature range from 94 to 153 °C. The enthalpy (H) calculated from the curves for the composites were recalculated only to refer to ABS matrix using the formula
(1)$$H_{\mathrm{ABS}}=\frac{H}{\Phi }$$

Taking into account the weight fraction (Φ) of ABS in the composite samples, it was found that copper only at amount 4.8 vol % (sample E) significantly affected the transitions in S-AN phases (Table 2).

These results may suggest that up to about 2.4 vol % of copper S-AN phase in ABS is not markedly affected by this filler during applied processing conditions. However, further increase (about 4.8 vol %) resulted in reduction of enthalpy from about roughly 7.8 J/g to about 5.9 J/g, showing in that case the influence of copper on S-AN structure.

The effect of copper content on oxidative degradation of ABS was evaluated by means of oxidation induction time (OIT) and oxidation induction temperature (OIT*). It was observed that during isothermal storing in oxygen atmosphere, OIT was determined to be about 12.2 min and this value was similar for all studied samples (Figure 3).

However, one can notice that copper, regardless of its contents, accelerated the dynamics of ABS degradation process. For instance, heat emitted from copper-containing samples reached 130 mW/g about 30 s earlier than it was attributed to neat ABS. Decreasing resistance to thermal oxidation with increasing content of copper was also noticed when the samples were heated from 50 to 280 °C in an oxygen atmosphere. The determined OIT* values were 203.4, 174.4, 157.2, 151.4, and 141.3 °C for the samples A, B, C, D, and E, respectively. The OIT* value for neat ABS is similar to that presented in literature, where OIT* values were dependent on injection moulding conditions and ranged from about 205 to 224 °C [33]. However, one should keep in mind that it was determined under different experimental conditions (under airflow, at rate of 100 mL/min, and for other grade of ABS) than performed in this work.

The TG analysis revealed that the values of the onset degradation temperature (T_On_) and the temperature (T_max_) at maximum rate of mass loss were similar for all studied samples; however, with increasing copper content the process ended at a clearly lower temperatures (T_End_), which indicates that even in a non-oxygen atmosphere copper can accelerate degradation process of ABS. The values of these temperatures for all tested samples are listed in Table 3 whereas selected TG/DTG curves are also shown in Figure 4.

It was also noted that DTG for neat ABS was clearly asymmetrical (two local maxima), while in the copper containing samples only one visible maximum (around 424 °C) could be perceived. This asymmetry can be explained by overlapping degradation of PB with SAN fractions. It is known that PB phase starts to degrade first followed by SAN phase, thus some asymmetry in the TG curve can observed, although these degradation processes are generally overlapped [35]. One can expect that copper filled samples, which better conduct the heat will undergo more dynamic degradation processes for both PB and SAN phases reducing this asymmetry for TG curves.

It is known that ABS subjected to heat can lose its impact strength [36]. However, a routine pendulum type of notched impact test generally does not yield significant differences in the impact values [37]. Therefore, in this study the energy (E_B_) provided to break the samples under static tension conditions (tension tests) were calculated using the formula
(2)$$E_{B}=\frac{1}{S}\int F\left(x\right)dx$$

As listed in Table 4, the values of E_B_ increased from about 16 to about 33 kJ/m^2^ after addition 0.6 vol % of Cu, whereas with further increase in Cu (4.8 vol %) it was reduced to about 15 kJ/m^2^.

The presence of copper proved to be not significantly detrimental on ABS resistance to be broken. Neat ABS had E_B_ of about 16 kJ/m^2^ and addition of small portion of Cu increased this value to about 33 kJ/m^2^ and then decreased along with increase in Cu content to about 15 kJ/m^2^. Higher copper content caused ABS to be stiffer and the Young modulus values were inversely correlated with E_B_. Melt flow rate of the samples decreased from about 27 to about 10 with increasing copper content. This significant reduction of MFR can impose higher processing temperatures in some manufacturing techniques (especially for thin wall injection moulded or 3D-printed products). On the other hand, higher processing temperature can lead to more intensive degradation of these composites as proved in this work.

The influence of copper on the volume resistivity of the composites was shown in Figure 5. The addition of copper reduced R_v_ from 2 × 10^16^ to 2 × 10^15^ Ω-cm, nearly regardless of its content. However, all composites characterized with good dielectric properties thus can be still used as isolators in various applications.

### 3.2. Effects of Infrared Lasers Irradiation and Electroless Metallization

The composite sample were irradiated with CO_2_ and Nd: YAG lasers to cause ablation of ABS matrix and thus uncover embedded Cu particles. It was expected that Cu particles will locally constitute the sites for reduction of copper from metallization bath to enable forming of continuous copper layer. Therefore, a broad range of parameters for laser irradiation was applied, then irradiated samples were metalized and tested by optical measurements to verify if the copper layer was successively deposited.

It was concluded that regardless of irradiation conditions CO_2_ laser was not able to activate the surface. However, by appropriate optimization of irradiation parameters, some composites were successfully metalized after irradiation with Nd:YAG laser. The best metallization effects (surface coverage with copper) was attained for composite E (4.8 vol % of Cu), however composite D was also partly metalized (Figure 6).

Based on the numerous irradiations at various parameters (power, scanning speed, and frequency) narrow laser processing window has been found for which composites E and D were successfully metalized. It was noticed that the most crucial irradiation parameter was the laser beam power which according to this study should be tailored at about 8 W. Then, the next important parameter was scanning speed of laser beam which has been established optimally from about 370 to about 410 mm/s. The effects of laser beam frequency on metallization seemed to be of less significance as compared with laser beam power and its scanning speed. Some examples of metallization effects for composites E and D irradiated at various preselected irradiation conditions at constant preselected power (P = 8 W) are presented in Figure 7.

As near (1064 nm) and far (10,6 µm) infrared lasers radiations have been applied to uncover embedded copper particles at least two heat induced phenomena can be expected. The first one is thermal degradation leading to ablation of polymer matrix which is desirable to uncover copper particles, while the second one, accompanying ablation, is melting/remelting of the polymer matrix and re-embedding copper particles. Laser-induced melting of the surface layer can be considered as detrimental for uncovering copper particles, and thus for surface activation. Mass losses of the composite E resulted from Nd:YAG and CO_2_ laser irradiations have been compared due to the importance of ablation process. The lasers were set to the same frequency (60 kHz), power (8 W), and scanning speed (410 mm/s) to evaluate the influence of lasers wavelengths on ablation rate of the composite E. It was found that irradiation with Nd:YAG laser caused about two times more mass loss (Δm = 2.6 ± 0.2 mg) as compared to that induced by CO_2_ laser (Δm = 1.1 ± 0.3 mg). Higher ablation intensity for Nd:YAG laser could be a crucial factor in finding suitable irradiation parameters leading to successful electroless metallization. Although IR radiation is well absorbed by ABS, it is well known that copper reflects well CO_2_ laser radiation, thus less of its beam power can be absorbed as compared with Nd:YAG laser. This feature could be significant for the different ablation characteristics induced by CO_2_ and Nd:YAG laser. It was additionally proved by cross-sectional views of the composite E irradiated with CO_2_ and Nd:YAG laser with the same irradiation parameters. As presented in Figure 8 CO_2_ laser affected significantly thinner surface layer (about 37 µm) than Nd:YAG laser (about 220 µm). 

As the samples irradiated with CO_2_ could not be metalized, regardless of irradiation conditions, further study was focus on the samples irradiated with Nd:YAG.

The composite samples irradiated with Nd:YAG laser differed in surface morphology (Figure 9). Small copper particles can be perceived on SEM images of the composites, however, their number increased with increasing copper content. Characteristic local degradation areas are clearly visible for composite B, which further reduced for composite C and almost disappeared for composites D and E. As discussed previously, PB phase is dispersed in continuous SAN phase and is more susceptible to and can initiate degradation process of ABS. Therefore, one can expect that observed degradation areas are representing laser ablated PB phase.

Along with increase of copper particles visible on the surface, its structure became more homogenously modified. It can result from the fact that with increase of copper content, the composites were probably more uniformly heated, and thus not only PB but also SAN phase were altogether degraded and this uniform surface temperature increase caused surface morphology features to become homogenized.

EDX analysis for composites A, B, and C revealed that dominant signals were derived from carbon and oxygen elements whereas copper were nearly not detected. However, for composites D and E copper emission band was detected and copper content estimated to be about 3.7 and 5.2 wt % for composites D and E, respectively. After metallization, 46.3 wt % and 74.7 wt % of copper were detected in composite D and E, respectively (Table 5).

Metallized surface of these composites are presented in Figure 10.

It is seen from Figure 10 that the composites surface was not fully covered with copper (it can be also perceived in Figure 6 and Figure 7). There are observed locally more dark areas which, in case of composite D, are significantly larger. They can represent ABS (no copper layer) or significantly thin copper layer. Nevertheless, metallized layers deposited on composites D and E were finely conductive (about 1 S/cm) as proved by standard multimeter, thus further electroplating is possible to be proceeded. The composites D and E were also tested as filaments in fused deposition modeling 3D printing process.

## 4. Conclusions

Application of spherical microparticles of copper as additive for ABS to laser-induce surface activation and electroless metallization was evaluated. The composites were manufactured by extrusion granulation and injection moulding. The effects of copper filler on structural properties of ABS was evaluated. It was found based on DSC analysis that copper only at amount 4.8 vol % (sample E) affected the transitions in S-AN phases, probably by accelerating thermal degradation during processing. However, oxidation induction time (OIT) was similar (about 12.2 min) for all studied samples. Once started, the isothermal oxidation process was accelerated by the presence of copper filler. The determined oxidation induction temperature (OIT*) values were reduced along with the increase in Cu content, which shows that the samples were more susceptible to temperature than time-induced thermal degradation. The presence of copper proved to be not significantly detrimental on ABS resistance to be broken. Break energy (E_B_) raised significantly after addition of 0.6 vol % Cu, then reduced along with higher Cu increase. Higher copper content caused ABS to be stiffer and the Young modulus values were inversely correlated with E_B_. The influence of copper on the volume resistivity of the composites was not significant and samples characterized with good dielectric properties. It was proved that regardless of irradiation conditions only Nd:YAG laser irradiation resulted in surface activation. Ablation rate and modified surface layer for CO_2_ was lower than for Nd:YAG using the same irradiation parameters (power, speed, and frequency of laser beams), thus the laser wavelength was crucial for successful surface activation. However, also for Nd:YAG irradiation parameters had to be precisely selected to obtain metallization effects. The best metallization coverage was obtained for samples E (4.8 vol %), however also in that case some copper layer discontinuities were locally detected. Nevertheless, electroless deposited copper was conductive, thus can serve as a base conductive material for subsequent electroplating process. The composite was ease to be processed by FDM 3D printing technique, thus can be used for rapid prototyping of MDI devices.

## Figures and Tables

**Figure 1 materials-13-02224-f001:**
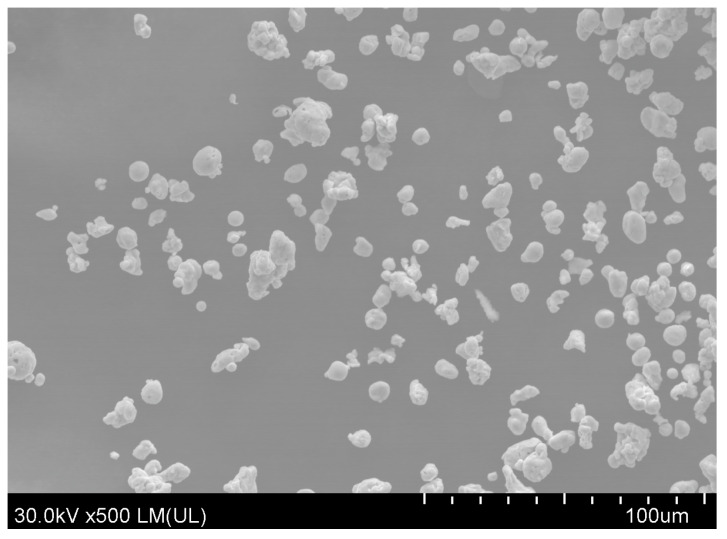
SEM image of copper particles used as a filler.

**Figure 2 materials-13-02224-f002:**
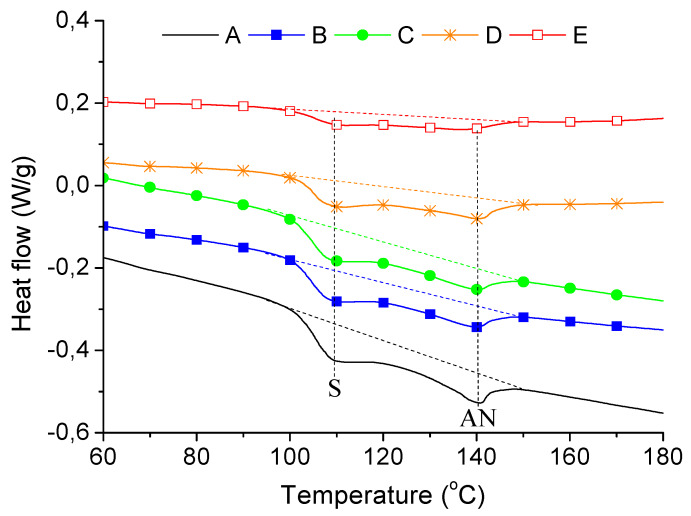
DSC curves for composites A, B, C, D, and E (second heating).

**Figure 3 materials-13-02224-f003:**
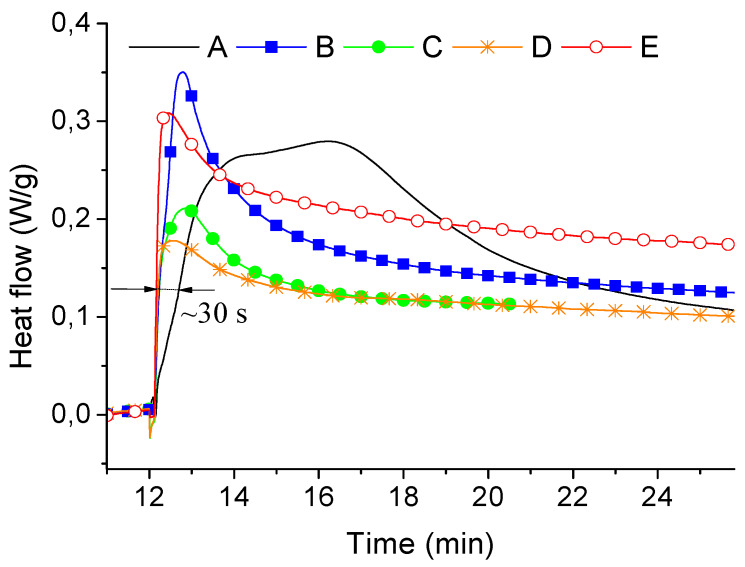
Oxidation induction time (isothermal OIT) for samples A, B, C, D, and E.

**Figure 4 materials-13-02224-f004:**
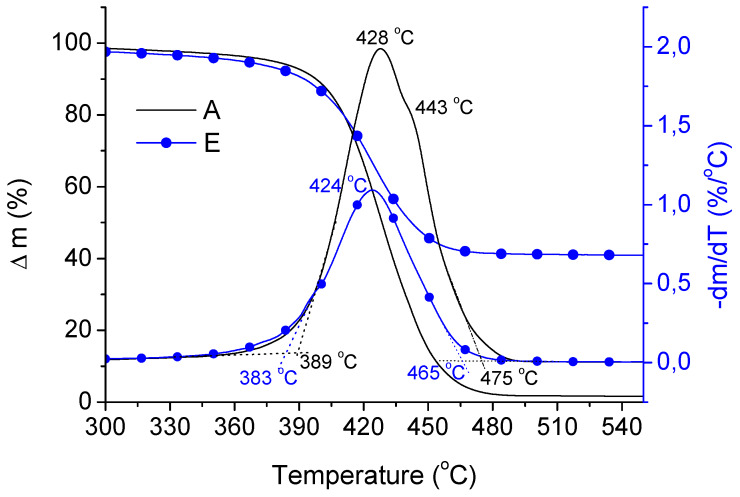
TG/DTG carves for A and E samples.

**Figure 5 materials-13-02224-f005:**
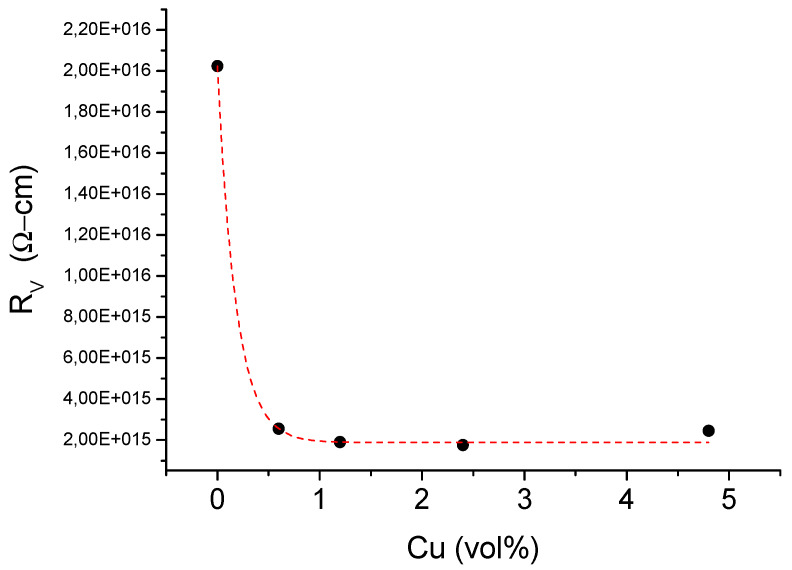
Volume resistivity (R_v_) versus copper content in ABS.

**Figure 6 materials-13-02224-f006:**
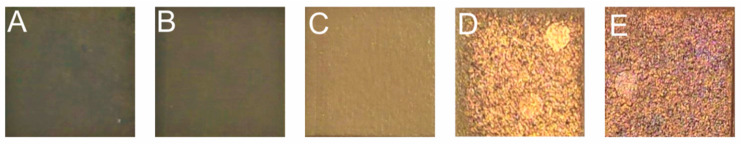
Effects of electroless metallization of the composites A, B, C, D, and E.

**Figure 7 materials-13-02224-f007:**
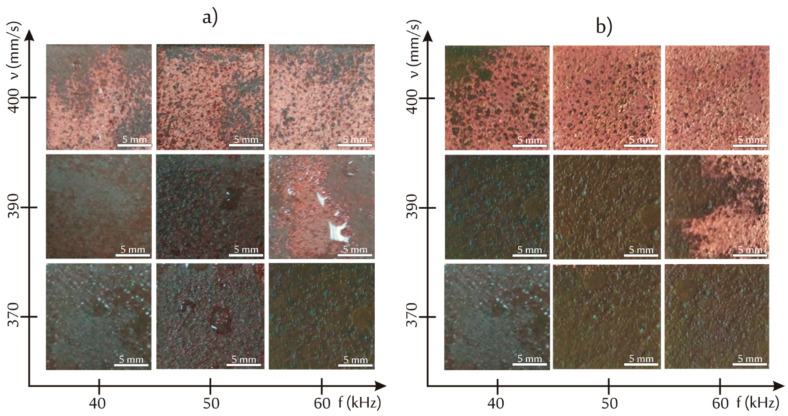
Optical images of the metalized composites: (**a**) D and (**b**) E irradiated at varied frequency and scanning speed of laser beam while maintaining constant power (P = 8 W).

**Figure 8 materials-13-02224-f008:**
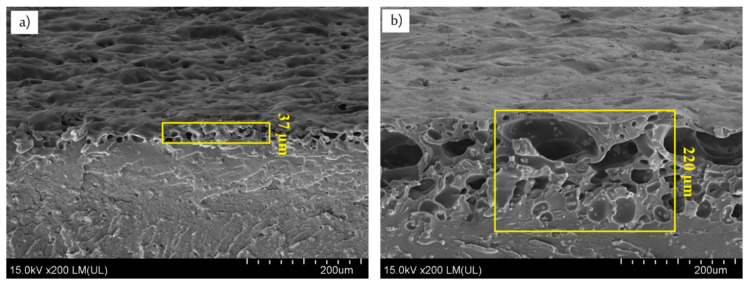
SEM images (side views) of composite E irradiated with (**a**) CO_2_ and (**b**) Nd:YAG lasers using the same processing parameters (P = 8 W, v = 410 mm/s, f = 60 kHz).

**Figure 9 materials-13-02224-f009:**
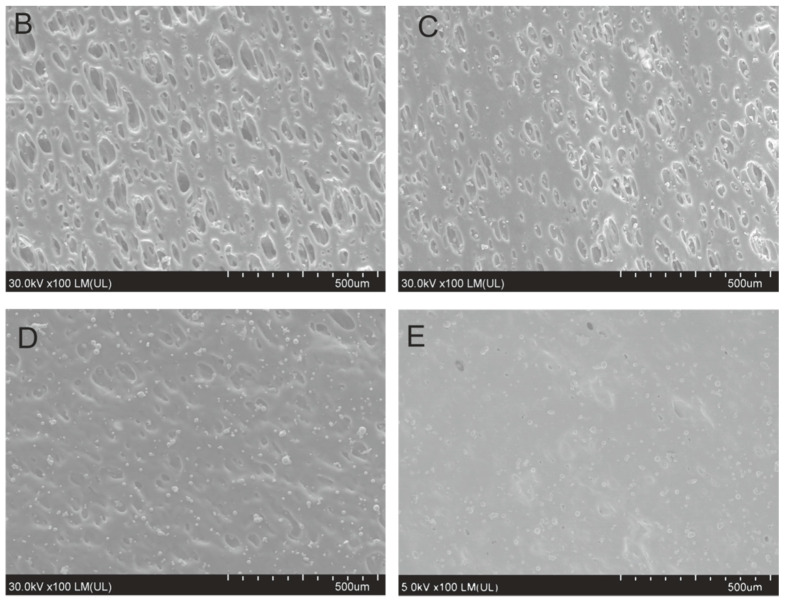
SEM images of the composites B, C, D, and E irradiated with Nd:YAG laser (P = 8 W, v = 410 mm/s, f = 60 kHz).

**Figure 10 materials-13-02224-f010:**
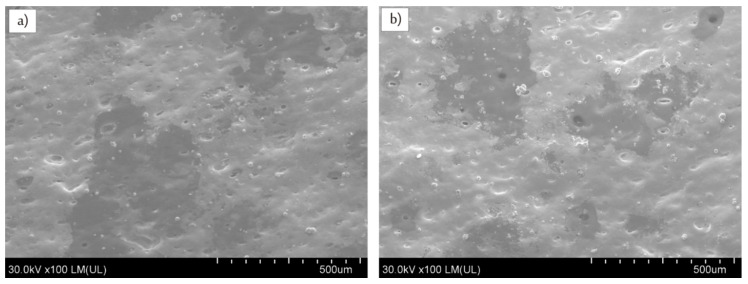
SEM images of composites: (**a**) D and (**b**) E, both irradiated and electroless metallized.

**Table 1 materials-13-02224-t001:** Designation of composite samples due to the copper content.

Samples	Cu (wt %)	Cu (vol %)
A	0	0
B	5	0.6
C	10	1.2
D	20	2.4
E	40	4.8

**Table 2 materials-13-02224-t002:** Weight fraction (Φ) of ABS in the composite samples (A/E) and enthalpies of composites (H) and of ABS matrix (H_ABS_)

Samples	Φ	H (J/g)	H_ABS_ (J/g)
A	1	8.4	8.4
B	0.95	7.5	7.9
C	0.9	7.3	8.1
D	0.8	6.2	7.8
E	0.6	3.53	5.9

**Table 3 materials-13-02224-t003:** Temperatures at onset (T_On_), end (T_End_) and maximum rate (T_max_) of mass loss determined based on DTG curves for studied samples (see Figure 4)

Samples	T_On_ (°C)	T_Max_ (°C)	T_End_ (°C)
A	389	428	475
B	382	428	466
C	385	429	467
D	386	429	466
E	383	424	465

**Table 4 materials-13-02224-t004:** Young modulus (E), tensile strength (σ_M_), strain at σ_M,_ (ε_M_), tensile at break (σ_B_), strain at σ_B_ (ε_B_), break energy (E_B_), melt flow rate (MFR) of the studied samples.

Sample	E (MPa)	σ_M_ (MPa)	ε_M_ (%)	σ_B_ (MPa)	ε_B_ (%)	E_B_ (kJ/m^2^)	MFR 220 °C; 5 kg
A	1143 ± 20	33.2 ± 1.9	4.93 ± 0.38	23.7 ± 5.0	7.0 ± 1.1	16.3 ± 1.6	26.5 ± 0.9
B	1082 ± 38	39.3 ± 1.7	6.80 ± 0.21	38.8 ± 1.6	7.0 ± 0.3	33.0 ± 2.5	21.9 ± 0.6
C	1103 ± 22	36.9 ± 1.8	5.42 ± 0.32	36.9 ± 1.8	5.4 ± 0.4	24.6 ± 2.4	16.5 ± 0.5
D	1276 ± 17	36.3 ± 1.6	4.80 ± 0.31	33.8 ± 3.9	5.2 ± 0.3	21.7 ± 1.5	12.0 ± 0.8
E	1398 ± 24	32.5 ±1.6	3.92 ± 0.25	30.2 ± 3.6	4.2 ± 0.3	14.9 ± 1.2	10.0 ± 0.5

**Table 5 materials-13-02224-t005:** EDX elemental analysis for composites E and D after laser irradiation (P = 8 W, v = 410 mm/s, f = 60 kHz) and metallization

Composite	Treatment	Element (wt %)		
		Cu	C	O
D	irradiated	3.7	96.3	0
D	metalized	46.3	39.3	14.2
E	irradiated	5.2	94.7	0
E	metalized	74.7	8.9	16.3

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
