# Peer review of "Copper Filled Poly(Acrylonitrile-co-Butadiene-co-Styrene) Composites for Laser-Assisted Selective Metallization"

_materials, 2020, doi:10.3390/ma13102224_

Round 1

Reviewer 1 Report

This manuscript presents the laser-assisted selective metallization on ABS composites incorporated with special additives of copper microparticles. The authors investigated two types of infrared lasers of 1064 nm Nd: YAG fiber laser and 10.6 μm CO2 laser for surface activation. The results showed that effective metallization could be realized by Nd: YAG fiber laser, while 10.6 μm CO2 laser was not. The experiments were carefully carried out, and the research results had a guiding role in the laser-assisted selective metallization industry.

However, the following issues and questions need to be addressed:

1. On page 2, line 43, “…MDI are more recyclable and their…” “MID” instead of “MDI,” please check it.

2. On page 2, line 68-70, “It can be predicated that copper particles will be less densely located on the surface as compared with nanoscopic metalorganic additives which are densely dispersed in polymer matrix.” Can the authors provide the corresponding experimental evidence or references?

3. In Section 2, line 82, “…1.04 g/cm3 and a flow rate of 55 cm3/10 min (10 kg, 220°C)…” “1.04 g/cm3” instead of “1.04 g/cm3”; the unit of flow rate is usually referring to “g/10 min” instead of “cm3/10 min”, and a unit conversion is needed.

4. On page 3, line 92, “…dried at 80º…” “80 ℃” instead of “80º.”

5. On page 4, line 152, “The elemental surface layer composition was were…”, delete “were.”

6. On page 5, line 183, “The curves shown in Figure 1 are obtained…” “Figure 2” instead of “Figure 1.”

7. On page 9, line 261, “…was applied, than irradiated…” “then” instead of “than.”

8. In Figure 7, the scale bar should be provided in optical images.

9. On page 10, line 290, “The lasers were set to the same frequency (60 Hz)…” “60 kHz” instead of “60 Hz.”

10. On page 11, line 322, “EDX analysis for composites…” where is the result of EDX analysis, please present it in this manuscript.

11. On page 12, line 334, “…composites D and E were finely conductive…”What is the specific conductivity?

12. The paper needs an extensive revision of English grammar.

In conclusion, this manuscript needs a minor revision before the publication in Materials.

Author Response

  1. On page 2, line 43, “…MDI are more recyclable and their…” “MID” instead of “MDI,” please check it.

It has been corrected

  1. On page 2, line 68-70, “It can be predicated that copper particles will be less densely located on the surface as compared with nanoscopic metalorganic additives which are densely dispersed in polymer matrix.” Can the authors provide the corresponding experimental evidence or references?

It is general expectation that at the same wt% content the larger are the particles the lower is their number per unit area of the surface (in that case the distance between the particles increases with their sizes).

  1. In Section 2, line 82, “…1.04 g/cm3 and a flow rate of 55 cm3/10 min (10 kg, 220°C)…” “1.04 g/cm3” instead of “1.04 g/cm3”; the unit of flow rate is usually referring to “g/10 min” instead of “cm3/10 min”, and a unit conversion is needed.

It has been corrected.

  1. On page 3, line 92, “…dried at 80º…” “80 ℃” instead of “80º.”

It has been corrected.

  1. On page 4, line 152, “The elemental surface layer composition was were…”, delete “were.”

It has been corrected.

  1. On page 5, line 183, “The curves shown in Figure 1 are obtained…” “Figure 2” instead of “Figure 1.”

It has been corrected.

  1. On page 9, line 261, “…was applied, than irradiated…” “then” instead of “than.”

It has been corrected.

  1. In Figure 7, the scale bar should be provided in optical images.

It has been incorporated.

  1. On page 10, line 290, “The lasers were set to the same frequency (60 Hz)…” “60 kHz” instead of “60 Hz.”

It has been corrected.

  1. On page 11, line 322, “EDX analysis for composites…” where is the result of EDX analysis, please present it in this manuscript.

A new table has been introduced.

  1. On page 12, line 334, “…composites D and E were finely conductive…”What is the specific conductivity?

It has been provided

  1. The paper needs an extensive revision of English grammar.

It has been performed.

In conclusion, this manuscript needs a minor revision before the publication in Materials.

I am thankful for all your well-founded remarks and suggestions, which helped me to improve the quality of this manuscript.

Reviewer 2 Report

In this study, the authors reported a manufacture technique to fabricate metallized products by injection molding of a copper/ABS composite, followed by a laser surface activation and subsequent electroless metallization. This is an interesting topic and the manuscript is well written. The copper/ABS composite and the metallized products were fully characterized, and various manufacturing parameters were also optimised. Therefore, I suggest accepting this manuscript for publication on MATERIALS after revision.

The author didn’t explain the experiment results (i.e DSC and TGA) in-depth. Specifically, why does sample E show a symmetrical DTG profile?

Line 101, "heating zones set to 220, 220, 215, 205 °C...". Should it be 225 oC? 

Author Response

The author didn’t explain the experiment results (i.e DSC and TGA) in-depth. Specifically, why does sample E show a symmetrical DTG profile?

Now, the discussion on symmetrical DTG profile has been provided and appropriate literature cited.

Line 101, "heating zones set to 220, 220, 215, 205 °C...". Should it be 225 oC?

It has been corrected

I am thankful for all the remarks, which helped me to improve the quality of this manuscript.

Reviewer 3 Report

The manuscript by Rytlewski et al. reports on the evaulation of microscopic copper particles as a metallization precursor for poly(acrylonitrile-butadiene-styrene). The topic of the manuscript is of interest for people woking in the materials science field, but it is an opinion of this reviewer that some revisions are required before futher processing. Below the main comments as per article sections:

Abstract

- In its present form it is too literal and too similar to an introduction section. Authors should revise it by inserting some key results of the study.

Introduction

- Authors should revise the section to better highlight the novelty of the study, e.g. by briefly explaining the main materials used in the broad literature review cited at lines 73-74.

Materials

- This reviewer is wondering if materials in paragraph 2.1 should be presented as bulleted list

- Please rename paragraph 2.2 to clearly highligh that the preparation method is reported there

Results and Discussion

- The presentation of the data is well organized, but the discussion part should be improved. As an example, authors should make comparison with different systems available in the lietrature to clearly prove the advantages of the proposed straetgy

References

- Authors reported groupes references in the introduction (Ref. 17-24). They should briefly discuss them, while un-needed ones (if any) should be removed.

Author Response

Abstract

  • In its present form it is too literal and too similar to an introduction section. Authors should revise it by inserting some key results of the study.

It has been corrected

Introduction

  • Authors should revise the section to better highlight the novelty of the study, e.g. by briefly explaining the main materials used in the broad literature review cited at lines 73-74.

There is in introduction section:

Due to the best authors’ knowledge and broad literature review this type of filler used as a metallization precursor for LDS technique was not the subject of previous publications. Additional novelty of this study is reflected by application of the two types of infrared lasers (1064 nm and 10.6 µm) for which activation effects were compared.

Materials

- This reviewer is wondering if materials in paragraph 2.1 should be presented as bulleted list

  • Please rename paragraph 2.2 to clearly highligh that the preparation method is reported there

It has been corrected

Results and Discussion

  • The presentation of the data is well organized, but the discussion part should be improved. As an example, authors should make comparison with different systems available in the lietrature to clearly prove the advantages of the proposed straetgy.

I didn’t find in literature similar systems (application of metallic particles as precursors) to compare with.

References

  • Authors reported groupes references in the introduction (Ref. 17-24). They should briefly discuss them, while un-needed ones (if any) should be removed.

I would like to leave these references as referencing to the very important issue of application metalorganic precursors for LDS technique. Interested readers can follow this articles to get better insight what was previously done, especially because of the lack of information for commercially available compounds for that technique.

Finally, I am thankful for all well-founded remarks and suggestions, which helped me to improve the quality of this manuscript.

Round 2

Reviewer 3 Report

Authors addressed all the reviewer's comments and provided suitable replies. The manuscript can be accepted for publication in Materials in its current form.